# Gemma Scope: Open Sparse Autoencoders Everywhere All At Once on Gemma 2

**Tom Lieberum, Senthooran Rajamanoharan, Arthur Conmy,**
**Lewis Smith, Nicolas Sonnerat, Vikrant Varma,**
**János Kramár, Anca Dragan, Rohin Shah, Neel Nanda**
Google DeepMind
tlieberum@google.com

## Abstract

Sparse autoencoders (SAEs) are an unsupervised method for learning a sparse decomposition of a neural network's latent representations into seemingly interpretable features. Despite recent excitement about their potential, research applications outside of industry are limited by the high cost of training a comprehensive suite of SAEs. In this work, we introduce Gemma Scope, an open suite of JumpReLU SAEs trained on all layers and sub-layers of Gemma 2 2B and 9B and select layers of Gemma 2 27B base models. We primarily train SAEs on the Gemma 2 pre-trained models, but additionally release SAEs trained on instruction-tuned Gemma 2 9B for comparison. We evaluate the quality of each SAE on standard metrics and release these results. We hope that by releasing these SAE weights, we can help make more ambitious safety and interpretability research easier for the community. Weights and a tutorial can be found at https://huggingface.co/google/gemma-scope and an interactive demo can be found at https://neuronpedia.org/gemma-scope.

## 1 Introduction

There are several lines of evidence that suggest that a significant fraction of the internal activations of language models are sparse, linear combination of vectors, each corresponding to meaningful features (Elhage et al., 2022; Gurnee et al., 2023; Olah et al., 2020; Park et al., 2023; Nanda et al., 2023a; Mikolov et al., 2013). But by default, it is difficult to identify which vectors are meaningful, or which meaningful vectors are present. Sparse autoencoders are a promising unsupervised approach to do this, and have been shown to often find causally relevant, interpretable directions (Bricken et al., 2023; Cunningham et al., 2023; Templeton et al., 2024; Gao et al., 2024; Marks et al., 2024). If this approach succeeds it could help unlock many of the hoped for applications of interpretability (Nanda, 2022; Olah, 2021; Hubinger, 2022), such as detecting and fixing hallucinations, being able to reliably explain and debug unexpected model behaviour and preventing deception or manipulation from autonomous AI agents.

However, sparse autoencoders are still an immature technique, and there are many open problems to be resolved (Templeton et al., 2024) before these downstream uses can be unlocked – especially validating or red-teaming SAEs as an approach, learning how to measure their performance, learning how to train SAEs at scale efficiently and well, and exploring how SAEs can be productively applied to real-world tasks.

As a result, there is an urgent need for further research, both in industry and in the broader community. However, unlike previous interpretability techniques like steering vectors (Turner et al., 2024; Li et al., 2023) or probing (Belinkov, 2022), sparse autoencoders can be highly expensive and difficult to train, limiting the ambition of interpretability research. Though there has been a lot of excellent work with sparse autoencoders on smaller models (Bricken et al., 2023; Cunningham et al., 2023; Dunefsky et al., 2024; Marks et al., 2024), the works that use SAEs on more modern models have normally focused on residual stream SAEs at a single layer (Templeton et al., 2024; Gao et al., 2024; Engels et al., 2024). In addition, many of these (Templeton et al., 2024; Gao et al., 2024) have been trained on proprietary models which makes it more challenging for the community at large to build on this work.

To address this we have trained and released the weights of Gemma Scope: a comprehensive, open suite of JumpReLU SAEs (Rajamanoharan et al., 2024b) on every layer and sublayer of Gemma 2 2B and 9B (Gemma Team, 2024a),[1] as well as select

---

[1]We also release one suite of transcoders (Dunefsky et al.

layers of the larger 27B model in this series. We release these weights under a permissive CC-BY-4.0 license[2] on HuggingFace to enable and accelerate research by other members of the research community.

Gemma Scope was a significant engineering challenge to train. It contains more than 400 sparse autoencoders in the main release[3], with more than 30 million learned features in total (though many features likely overlap), trained on 4-16B tokens of text each. We used over 20% of the training compute of GPT-3 (Brown et al., 2020), saved about 20 Pebibytes (PiB) of activations to disk, and produced hundreds of billions of sparse autoencoder parameters in total. This was made more challenging by our decision to make a *comprehensive* suite of SAEs, on every layer and sublayer. We believe that a comprehensive suite is essential for enabling more ambitious applications of interpretability, such as circuit analysis (Conmy et al., 2023; Wang et al., 2022; Hanna et al., 2023), essentially scaling up Marks et al. (2024) to larger models, which may be necessary to answer mysteries about larger models like what happens during chain of thought or in-context learning.

In Section 2 we provide background on SAEs in general and JumpReLU SAEs in particular. Section 3 contains details of our training procedure, hyperparameters and computational infrastructure. We run extensive evaluations on the trained SAEs in Section 4.

## 2 Preliminaries

### 2.1 Sparse autoencoders

Given activations $\mathbf{x} \in \mathbb{R}^n$ from a language model, a sparse autoencoder (SAE) decomposes and reconstructs the activations using a pair of encoder and decoder functions $(\mathbf{f}, \hat{\mathbf{x}})$ defined by:

$$\mathbf{f}(\mathbf{x}) := \sigma\left(\mathbf{W}_{\text{enc}}\mathbf{x} + \mathbf{b}_{\text{enc}}\right), \qquad (1)$$

$$\hat{\mathbf{x}}(\mathbf{f}) := \mathbf{W}_{\text{dec}}\mathbf{f} + \mathbf{b}_{\text{dec}}. \qquad (2)$$

These functions are trained to map $\hat{\mathbf{x}}(\mathbf{f}(\mathbf{x}))$ back to $\mathbf{x}$, making them an autoencoder. Thus, $\mathbf{f}(\mathbf{x}) \in \mathbb{R}^M$

is a set of linear weights that specify how to combine the $M \gg n$ columns of $\mathbf{W}_{\text{dec}}$ to reproduce $\mathbf{x}$. The columns of $\mathbf{W}_{\text{dec}}$, which we denote by $\mathbf{d}_i$ for $i = 1 \ldots M$, represent the dictionary of directions into which the SAE decomposes $\mathbf{x}$. We will refer to to these learned directions as *latents* to disambiguate between learnt 'features' and the conceptual features which are hypothesized to comprise the language model's representation vectors.[4]

The decomposition $\mathbf{f}(\mathbf{x})$ is made *non-negative* and *sparse* through the choice of activation function $\sigma$ and appropriate regularization, such that $\mathbf{f}(\mathbf{x})$ typically has much fewer than $n$ non-zero entries. Initial work (Cunningham et al., 2023; Bricken et al., 2023) used a ReLU activation function to enforce non-negativity, and an L1 penalty on the decomposition $\mathbf{f}(\mathbf{x})$ to encourage sparsity. TopK SAEs (Gao et al., 2024) enforce sparsity by zeroing all but the top K entries of $\mathbf{f}(\mathbf{x})$, whereas the JumpReLU SAEs (Rajamanoharan et al., 2024b) enforce sparsity by zeroing out all entries of $\mathbf{f}(\mathbf{x})$ below a positive threshold. Both TopK and JumpReLU SAEs allow for greater separation between the tasks of determining which latents are active, and estimating their magnitudes.

### 2.2 JumpReLU SAEs

In this work we focus on JumpReLU SAEs as they have been shown to be a slight Pareto improvement over other approaches, and allow for a variable number of active latents at different tokens (unlike TopK SAEs).

**JumpReLU activation** The JumpReLU activation is a shifted Heaviside step function as a gating mechanism together with a conventional ReLU:

$$\sigma(\mathbf{z}) = \text{JumpReLU}_{\boldsymbol{\theta}}(\mathbf{z}) := \mathbf{z} \odot H(\mathbf{z} - \boldsymbol{\theta}). \quad (3)$$

Here, $\boldsymbol{\theta} > 0$ is the JumpReLU's vector-valued learnable threshold parameter, $\odot$ denotes elementwise multiplication, and $H$ is the Heaviside step function, which is 1 if its input is positive and 0 otherwise. Intuitively, the JumpReLU leaves the pre-activations unchanged above the threshold, but sets them to zero below the threshold, with a different learned threshold per latent.

**Loss function** As loss function we use a squared error reconstruction loss, and directly regularize

---

(2024); Appendix C), a 'feature-splitting' suite of SAEs with multiple widths trained on the same site (Section 4.2), and some SAEs trained on the Gemma 2 9B IT model (Kissane et al. (2024b); Section 4.4).

[2] Note that the Gemma 2 models are released under a different, custom license.

[3] For each model, layer and site we in fact release multiple SAEs with differing levels of sparsity; taking this into account, we release the weights of over 2,000 SAEs in total.

[4] This is different terminology from earlier work (Bricken et al., 2023; Rajamanoharan et al., 2024a,b), where feature is normally used interchangeably for both SAE latents and the language models features

the number of active (non-zero) latents using the L0 penalty:

$$\mathcal{L} := \|\mathbf{x} - \hat{\mathbf{x}}(\mathbf{f}(\mathbf{x}))\|_2^2 + \lambda \|\mathbf{f}(\mathbf{x})\|_0, \qquad (4)$$

where $\lambda$ is the sparsity penalty coefficient. Since the L0 penalty and JumpReLU activation function are piecewise constant with respect to threshold parameters $\boldsymbol{\theta}$, we use straight-through estimators (STEs) to train $\boldsymbol{\theta}$, using the approach described in Rajamanoharan et al. (2024b). This introduces an additional hyperparameter, the kernel density estimator bandwidth $\varepsilon$, which controls the quality of the gradient estimates used to train the threshold parameters $\boldsymbol{\theta}$.[5]

## 3 Training details

### 3.1 Data

We train SAEs on the activations of Gemma 2 models generated using text data from the same distribution as the pretraining text data for Gemma 1 (Gemma Team, 2024b) , except for the one suite of SAEs trained on the instruction-tuned (IT) model (Section 4.4). We generate activations on sequences of length 1024.

For a given sequence we only collect activations from tokens which are neither BOS, EOS, nor padding. After activations have been generated, they are shuffled in buckets of about $10^6$ activations. We speculate that a perfect shuffle would not significantly improve results, but this was not systematically checked. We would welcome further investigation into this topic in future work.

During training, activation vectors are normalized by a fixed scalar to have unit mean squared norm.[6] This allows more reliable transfer of hyperparameters (in particular the sparsity coefficient $\lambda$ and bandwidth $\varepsilon$) between layers and sites, as the raw activation norms can vary over multiple orders of magnitude, changing the scale of the reconstruction loss in Eq. (4). Once training is complete, we rescale the trained SAE parameters so that no

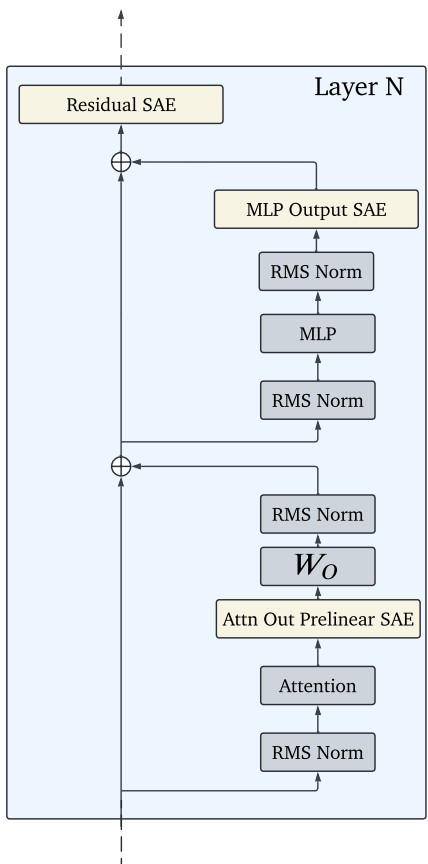

Figure 1: Locations of sparse autoencoders inside a transformer block of Gemma 2. Note that Gemma 2 has RMS Norm at the start and end of each attention and MLP block.

input normalization is required for inference (see Appendix B for details).

As shown in Table 1, SAEs with 16.4K latents are trained for 4B tokens, while 1M-width SAEs are trained for 16B tokens. All other SAEs are trained for 8B tokens.

**Location** We train SAEs on three locations per layer, as indicated by Fig. 1. We train on the attention head outputs before the final linear transformation $W_O$ and RMSNorm has been applied (Kissane et al., 2024a), on the MLP outputs after the RMSNorm has been applied and on the post MLP residual stream. For the attention output SAEs, we concatenate the outputs of the individual attention heads and learn a joint SAE for the full set of heads. We zero-index the layers, so layer 0 refers to the first transformer block after the embedding layer. In Appendix C we define *transcoders* (Dunefsky et al., 2024) and train one suite of these.

---

[5] A large value of $\varepsilon$ results in biased but low variance estimates, leading to SAEs with good sparsity but sub-optimal fidelity, whereas a low value of $\varepsilon$ results in high variance estimates that cause the threshold to fail to train at all, resulting in SAEs that fail to be sparse. We find through hyperparameter sweeps across multiple layers and sites that $\varepsilon = 0.001$ provides a good trade-off (when SAE inputs are normalized to have an unit mean squared norm) and use this to train the SAEs released as part of Gemma Scope.

[6] This is similar in spirit to Conerly et al. (2024), who normalize the dataset to have mean norm of $\sqrt{d_{\text{model}}}$.

## 3.2 Hyperparameters

**Optimization** We use the same bandwidth $\varepsilon = 0.001$ and learning rate $\eta = 7 \times 10^{-5}$ across all training runs. We use a cosine learning rate warmup from $0.1\eta$ to $\eta$ over the first 1,000 training steps. We train with the Adam optimizer (Kingma and Ba, 2017) with $(\beta_1, \beta_2) = (0, 0.999)$, $\epsilon = 10^{-8}$ and a batch size of 4,096. We use a linear warmup for the sparsity coefficient from 0 to $\lambda$ over the first 10,000 training steps.

During training, we parameterise the SAE using a pre-encoder bias (Bricken et al., 2023), subtracting $\mathbf{b}_{\mathrm{dec}}$ from activations before the encoder. However, after training is complete, we fold this bias into the encoder parameters, so that no pre-encoder bias needs to be applied during inference. See Appendix B for details.

Throughout training, we restrict the columns of $\mathbf{W}_{\mathrm{dec}}$ to have unit norm by renormalizing after every update. We also project out the part of the gradients parallel to these columns before computing the Adam update, as described in Bricken et al. (2023).

**Initialization** We initialize the JumpReLU threshold as the vector $\boldsymbol{\theta} = \{0.001\}^M$. We initialize $\mathbf{W}_{\mathrm{dec}}$ using He-uniform (He et al., 2015) initialization and rescale each latent vector to be unit norm. $\mathbf{W}_{\mathrm{enc}}$ is initalized as the transpose of $\mathbf{W}_{\mathrm{dec}}$, but they are not tied afterwards (Conerly et al., 2024; Gao et al., 2024). The biases $\mathbf{b}_{\mathrm{dec}}$ and $\mathbf{b}_{\mathrm{enc}}$ are initialized to zero vectors.

## 3.3 Infrastructure

### 3.3.1 Accelerators

**Topology** We train most of our SAEs using TPUv3 in a 4x2 configuration. Some SAEs, especially the most wide ones, were trained using TPUv5p in either a 2x2x1 or 2x2x4 configuration.

**Sharding** We train SAEs with 16.4K latents with maximum amount of data parallelism, while using maximal amounts of tensor parallelism using Megatron sharding (Shoeybi et al., 2020) for all other configurations. We find that as one goes to small SAEs and correspondingly small update step time, the time spent on host-to-device (H2D) transfers outgrows the time spent on the update step, favoring data sharding. For larger SAEs on the other hand, larger batch sizes enable higher arithmetic intensity by reducing transfers between HBM and VMEM of the TPU. Furthermore, the specific architecture of SAEs means that when using Megatron sharding, device-to-device (D2D) communication is minimal, while data parallelism causes a costly all-reduce of the full gradients. Thus we recommend choosing the smallest degree of data sharding such that the H2D transfer takes slightly less time than the update step.

As an example, with proper step time optimization this enables us to process one batch for a 131K-width SAE in 45ms on 8 TPUv3 chips, i.e. a model FLOP utilization (MFU) of about 50.8%.

### 3.3.2 Data Pipeline

**Disk storage** We store all collected activations on hard drives as raw bytes in shards of 10-20GiB. We use 32-bit precision in all our experiments. This means that storing 8B worth of activations for a single site and layer takes about 100TiB for Gemma 2 9B, or about 17PiB for all sites and layers of both Gemma 2 2B and 9B. The total amount is somewhat higher still, as we train some SAEs for 16B tokens and also train some SAEs on Gemma 2 27B, as well as having a generous buffer of additional tokens. While this is a significant amount of disk space, it is still cheaper than regenerating the data every time one wishes to train an SAE on it. Concretely, in our calculations we find that storing activations for 10-100 days is typically at least an order of magnitude cheaper than regenerating them one additional time. The exact numbers depend on the model used and the specifics of the infrastructure, but we expect this relationship to hold true in general. If there is a hard limit on the amount of disk space available, however, or if fast disk I/O can not be provided (see next paragraph), then this will favor on-the-fly generation instead. This would also be the case if the exact hyperparameter combinations were known in advance. In practice, we find it advantageous for research iteration speed to be able to sweep sparsity independently from other hyperparameters and to retrain SAEs at will.

**Disk reads** Since SAEs are very shallow models with short training step times and we train them on activation vectors rather than integer-valued tokens, training them requires high data throughput. For instance, to train a single SAE on Gemma 2 9B without being bottlenecked by data loading requires more than 1 GiB/s of disk read speed. This demand is further amplified when training multiple SAEs on the same site and layer, e.g. with different sparsity coefficients, or while conducting hyperparameter

| Gemma 2 Model | SAE Width | Attention | MLP | Residual | # Tokens |
|---|---|---|---|---|---|
| 2.6B PT | $2^{14} \approx 16.4K$ | **All** | **All** | **All+** | 4B |
| (26 layers) | $2^{15}$ | ✗ | ✗ | {12} | 8B |
| | $2^{16}$ | **All** | **All** | **All** | 8B |
| | $2^{17}$ | ✗ | ✗ | {12} | 8B |
| | $2^{18}$ | ✗ | ✗ | {12} | 8B |
| | $2^{19}$ | ✗ | ✗ | {12} | 8B |
| | $2^{20} \approx 1M$ | ✗ | ✗ | {5, 12, 19} | 16B |
| 9B PT | $2^{14}$ | **All** | **All** | **All** | 4B |
| (42 layers) | $2^{15}$ | ✗ | ✗ | {20} | 8B |
| | $2^{16}$ | ✗ | ✗ | {20} | 8B |
| | $2^{17}$ | **All** | **All** | **All** | 8B |
| | $2^{18}$ | ✗ | ✗ | {20} | 8B |
| | $2^{19}$ | ✗ | ✗ | {20} | 8B |
| | $2^{20}$ | ✗ | ✗ | {9, 20, 31} | 16B |
| 27B PT (46 layers) | $2^{17}$ | ✗ | ✗ | {10, 22, 34} | 8B |
| 9B IT | $2^{14}$ | ✗ | ✗ | {9, 20, 31} | 4B |
| (42 layers) | $2^{17}$ | ✗ | ✗ | {9, 20, 31} | 8B |

Table 1: Overview of the SAEs that were trained for which sites and layers. For each model, width, site and layer, we release multiple SAEs with differing levels of sparsity (L0).
**All+**: We also train one suite of transcoders on the MLP sublayers on Gemma 2.6B PT (Appendix C).

tuning.

To overcome this bottleneck we implement a shared server system, enabling us to amortize disk reads for a single site and layer combination:

- **Shared data buffer:** Multiple training jobs share access to a single server. The server maintains a buffer containing a predefined number of data batches. Trainers request these batches from the servers as needed.
- **Distributed disk reads:** To enable parallel disk reads, we deploy multiple servers for each site and layer, with each server exclusively responsible for a contiguous slice of the data.
- **Dynamic data fetching:** As trainers request batches, the server continually fetches new data from the dataset, replacing the oldest data within their buffer.
- **Handling speed differences:** To accommodate variations in trainer speeds caused by factors like preemption, crashes and different SAE widths, trainers keep track of the batches they have already processed. If a trainer lags behind, the servers can loop through the dataset again, providing the missed batches. Note that different training speeds result in different trainers not seeing the same data or necessarily in the same order. In practice we

found this trade-off well worth the efficiency gains.

## 4 Evaluation

In this section we evaluate the trained SAEs from various different angles. We note however that as of now there is no consensus on what constitutes a reliable metric for the quality of a sparse autoencoder or its learned latents and that this is an ongoing area of research and debate (Gao et al., 2024; Karvonen et al., 2024; Makelov et al., 2024).

Unless otherwise noted all evaluations are on sequences from the same distribution as the SAE training data, i.e. the pretraining distribution of Gemma 1 (Gemma Team, 2024b).

### 4.1 Evaluating the sparsity-fidelity trade-off

**Methodology** For a fixed dictionary size, we trained SAEs of varying levels of sparsity by sweeping the sparsity coefficient $\lambda$. We then plot curves showing the level of reconstruction fidelity attainable at a given level of sparsity.

**Metrics** We use the mean L0-norm of latent activations, $\mathbb{E}_{\mathbf{x}}\|\mathbf{f}(\mathbf{x})\|_0$, as a measure of sparsity. To measure reconstruction fidelity, we use two metrics:

- Our primary metric is delta LM loss, the increase in the cross-entropy loss experienced

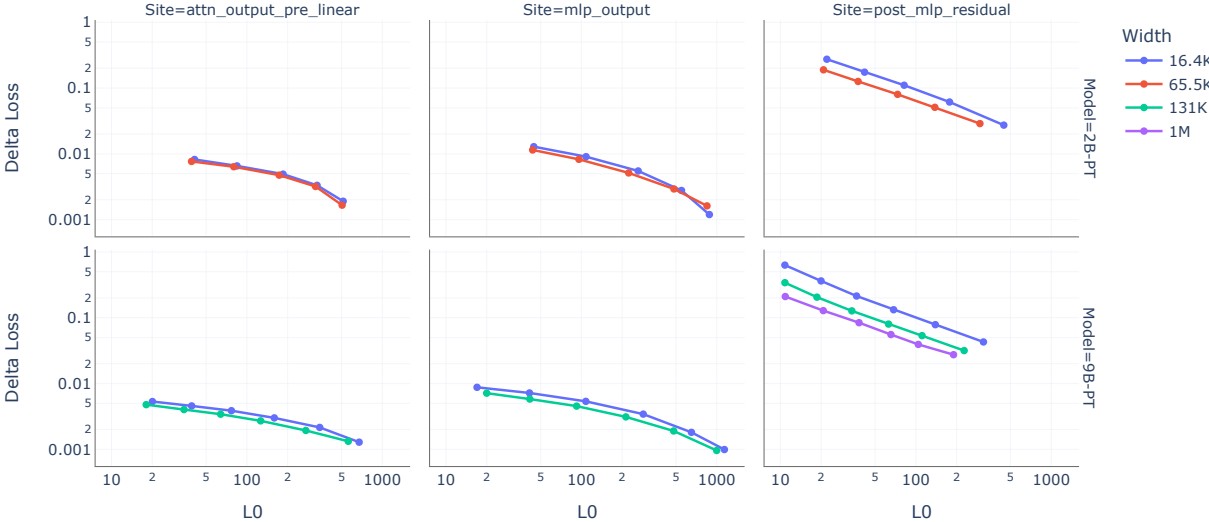

Figure 2: Sparsity-fidelity trade-off for layer 12 Gemma 2 2B and layer 20 Gemma 2 9B SAEs. An ideal SAE should have low delta loss and low L0, i.e. correspond to a point towards the bottom-left corner of each plot. For an analogous plot using FVU as the measure of fidelity see Fig. 11.

by the LM when we splice the SAE into the LM's forward pass.

- As a secondary metric, we also use fraction of variance unexplained (FVU) – also called the normalized loss (Gao et al., 2024) – as a measure of reconstruction fidelity. This is the mean reconstruction loss $\mathcal{L}_{\text{reconstruct}}$ of a SAE normalized by the reconstruction loss obtained by always predicting the dataset mean. Note that FVU is purely a measure of the SAE's ability to reconstruction the input activations, not taking into account the causal effect of any error on the downstream loss.

All metrics were computed on 2,048 sequences of length 1,024, after masking out BOS, EOS, and padding tokens when aggregating the results.

**Results** Fig. 2 compares the sparsity-fidelity trade-off for SAEs in the middle of each Gemma model. For the full results see Appendix D. Delta loss is consistently higher for residual stream SAEs compared to MLP and attention SAEs, whereas FVU (Fig. 11) is roughly comparable across sites. We conjecture this is because even small errors in reconstructing the residual stream can have a significant impact on LM loss, whereas relatively large errors in reconstructing a single MLP or attention sub-layer's output have a limited impact on the LM loss.[7]

## 4.2 Studying the effect of SAE width

Holding all else equal, wide SAEs learn more latent directions and provide better reconstruction fidelity at a given level of sparsity than narrow SAEs. Intuitively, this suggests that we should use the widest SAEs practicable for downstream tasks. However, there are also signs that this intuition may come with caveats. The phenomenon of 'feature-splitting' (Bricken et al., 2023) – where latents in a narrow SAE seem to split into multiple specialized latents within wider SAEs – is one sign that wide SAEs do not always use their extra capacity to learn a greater breadth of features (Bussmann et al., 2024). It is plausible that the sparsity penalty used to train SAEs encourages wide SAEs to learn frequent compositions of existing features instead of or in addition to learning new features (Anders et al., 2024). If this is the case, it is currently unclear whether this is good or bad for the usefulness of SAEs on downstream tasks.

In order to facilitate research into how SAEs' properties vary with width, and in particular how SAEs with different widths trained on the same data relate to one another, we train and release a 'feature-splitting' suite of mid-network residual stream SAEs for Gemma 2 2B and 9B PT with matching sparsity coefficients and widths between $2^{14} \approx 16.4\text{K}$ and $2^{19} \approx 524\text{K}$ in steps of powers of

---

[7]The residual stream is the bottleneck by which the previous layers communicate with all later layers. Any given MLP or attention layer adds to the residual stream, and is typically only a small fraction of the residual, so even a large error in the layer is a small error to the residual stream and so to the

rest of the network's processing. On the other hand, a large error to the residual stream itself will significantly harm loss. For the same reason, mean ablating the residual stream has far higher impact on the loss than mean ablating a single layer.

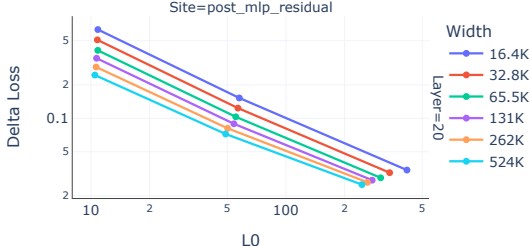

Figure 3: Delta loss versus sparsity curves for a series of SAEs of differing width (keeping $\lambda$ and other hyperparameters constant), trained on the residual stream after layer 20 of Gemma 2 9B.

two.[8] The SAEs are trained with different sparsity settings after layers 12 and 20 of Gemma 2 2B and 9B respectively.

**Sparsity-fidelity trade-off** Similar to Section 4.1, Fig. 3 compares fidelity-versus-sparsity curves for SAEs of differing width in this ladder.

**Latent firing frequency** Fig. 4 shows frequency histograms for $\lambda = 6 \times 10^{-4}$ SAEs in the same ladder of widths from $2^{14}$ to $2^{19}$ latents. To compute these, we calculate the firing frequency of each latent over 20,000 sequences of length 1,024, masking out special tokens. The mode and most of the mass shifts towards lower frequencies with increased number of latents. However there remains a cluster of ultra-high frequency latents, which has also been observed for TopK SAEs but not for Gated SAEs (Cunningham and Conerly, 2024; Gao et al., 2024; Rajamanoharan et al., 2024b).

### 4.3 Interpretability of latents

The interpretability of latents for JumpReLU SAEs and other architectures was investigated in Rajamanoharan et al. (2024b), finding little difference between various SAE architectures. Since we also use JumpReLU SAEs, we refer to section 5.3 of that work for a detailed discussion of the methodology and results.

### 4.4 SAEs trained on base models transfer to IT models

**Additional IT SAE training** Prior research has shown that SAEs trained on base model activations also faithfully reconstruct the activations of IT models derived from these base models (Kissane et al., 2024b). We find further evidence for these results

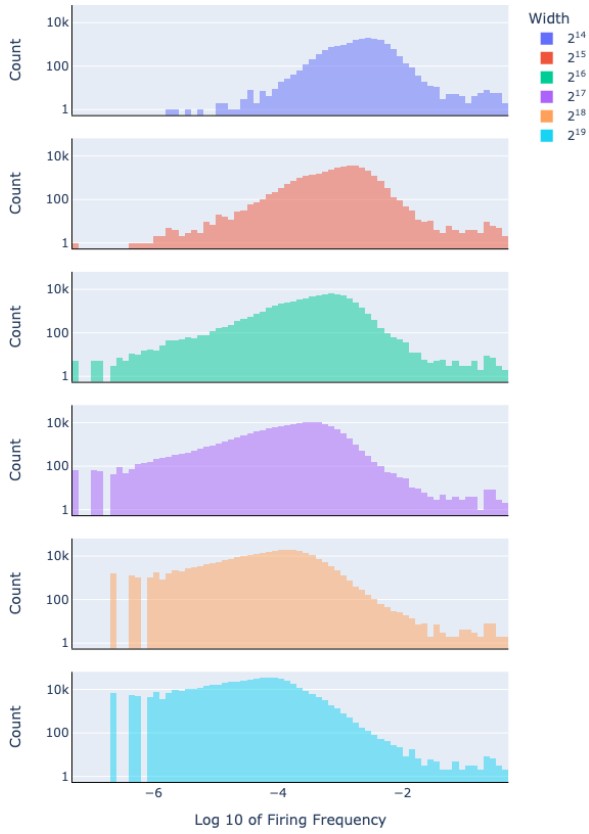

Figure 4: Frequency histogram of SAEs trained on Gemma 2 9B, layer 20, post MLP residual with sparsity coefficient $\lambda = 6 \times 10^{-4}$. (These correspond to the SAEs with L0 $\approx 50$ in Fig. 3.)

by comparing the Gemma Scope SAEs with several SAEs we train on the activations from Gemma 2B 9B IT. Specifically, we train these IT SAEs by taking the same pretraining documents used for all other SAEs (Section 3.1) and prepend them with Gemma's IT prefix for the user's query, and append Gemma's IT prefix for the model's response.[9] We then train each SAE to reconstruct activations at all token positions besides the user prefix (since these tokens have much larger norm (Kissane et al., 2024b), and are the same for every document). We also release the weights for these SAEs to enable further research into the differences between training SAEs on base and IT models. [10]

**Methodology** To evaluate the SAEs trained on the IT model's activations, we generate 1,024 rollouts of the Gemma 2 9B IT model on a random sample of the SFT data used to train Gemini 1.0 Ultra (GoogleDeepmind, 2024) , with temperature 1.0.

---

[8]Note the 1M-width SAEs included in Fig. 2 do not form part of this suite as they were trained using a different range of values for the sparsity coefficient $\lambda$.

[9]See e.g. `https://huggingface.co/google/gemma-2-2b-it` for the user and model prefixes.

[10]`https://huggingface.co/google/gemma-scope-9b-it-res`

We then use SAEs trained on the residual stream of the base model and the IT model to reconstruct these activations, and measured the FVU.

**Results** In Fig. 5 we show that using PT model SAEs results in increases in cross-entropy loss almost as small as the increase from the SAEs directly trained on the IT model's activation. We show further evaluations such as on Gemma 2 2B, measuring FVU rather than loss, and using activations from the user query (not just the rollout) in Appendix D.6. In Fig. 19 we find that the FVU for the PT SAEs is somewhat faithful, but does not paint as strong a picture as Fig. 5. A speculative explanation for this is that finetuning consists of 're-weighting' old features from the base model, in addition to learning some new, chat-specific features that do not have as big an impact on next-token prediction. This would mean the FVU looks worse than the increase in loss since the FVU would be impacted by low impact chat features, but change in loss would not be.

Future work could look into finetuning these SAEs on chat interactions if even lower reconstruction error is desired (Kissane et al., 2024b), or evaluating on multi-turn and targeted rollouts.

### 4.5 Additional evaluation results

In Appendix D, we present additional evaluation results covering

- Sparsity-Fidelity trade-off for more SAEs

- Studying SAE performance as a function of token position.

- SAE performance on different subsets of The Pile (Gao et al., 2020), showing stronger performance on Deepmind Mathematics (Saxton et al., 2019) and weaker performance on Europarl (Koehn, 2005).

- Impact of low precision inference, showing little performance regression from using `bfloat16`.

- Uniformity of active latent importance, which is a measure for how diffuse the downstream effect of a single SAE latent is, introduced by Rajamanoharan et al. (2024b).

- Additional evaluation results for SAEs trained on the activations of Gemma 2 IT models.

## 5 Related Work

**Open Weights Sparse Autoencoders** There have been several open weights SAE contributions by the research community. However, all releases we are aware of have focused on smaller and older language models or have not released a comprehensive set of autoencoder weights.

Marks and Mueller (2023) trained SAEs on the MLP outputs of all layers of Pythia-70M (Biderman et al., 2023). Braun et al. (2024) trained different variants of SAEs on GPT-2 small (Radford et al., 2019) and Tinystories-1M (Eldan and Li, 2023) on the residual stream activations in select layers. Belrose (2024) released TopK SAEs on the residual stream of Llama 3.1 8B (Meta, 2024). Engels et al. (2024) released Mistral 7B SAEs trained on the residual stream in layers 8, 16, and 24. Gao et al. (2024) released various SAEs on GPT-2 small with the latest release including TopK SAEs on every layer and sublayer, including the post-attention residual stream. Kissane et al. (2024c) released SAEs on the attention output of every layer of GPT-2 small. Kissane et al. (2024b) released PT, IT, and fine-tuned SAEs for Mistral-7B and Qwen 1.5 0.5B on the residual stream in the middle of the language model. Dunefsky et al. (2024) released MLP transcoders on all layers of GPT-2 small. Han (2024) released an SAE on the residual stream of layer 25 of Llama 3B IT. We also refer to f SAEs supported by the SAELens (Joseph Bloom, 2024) library for an overview of easily accessible open weights SAEs .

In contrast to the above work, Gemma Scope is the first release of SAE weights which contains SAEs for all layers and sublayers of a recently released, performant 2B and 9B language model.

## 6 Discussion and Future Work

In this report we have introduced Gemma Scope, a comprehensive suite of Sparse Autoencoders (SAEs) on all layers and sublayers of Gemma 2 2B and 9B PT. We have described the engineering challenges involved in this project and how we approached them. In order to shed light on the quality of the Gemma Scope SAEs, we have provided results of various evaluation experiments. While we have extensively evaluated these SAEs, their real test is how much they can enable and accelerate downstream interpretability research. To further underscore this point, we provide a broad range of open research questions related to SAEs which

we think are enabled or aided by Gemma Scope in Appendix A and which we would be excited to see pursued by the interpretability research community.

Training such a comprehensive suite of Sparse Autoencoders requires a significant upfront cost in compute and energy (Section 3) and thus also has a certain carbon footprint. It is our hope that by paying this cost once, we can avoid the broader research community having to train their own models again and again. We think Gemma Scope will enable research into language model internals for years to come, even if and when the state of the art of SAE training improves in the future, and so we are optimistic that the cost of training Gemma Scope can be amortized.

## Acknowledgements

We are incredibly grateful to Joseph Bloom, Johnny Lin and Curt Tigges for their help creating an interactive demo of Gemma Scope on Neuronpedia (Lin and Bloom, 2023), creating tooling for researchers like feature dashboards, and help making educational materials. We are grateful to Alex Tomala for engineering support and Tobi Ijitoye for organizational support during this project. Additionally, we would like to thank Meg Risdal, Kathleen Kenealy, Joe Fernandez, Kat Black and Tris Warkentin for support with integration with Gemma, and Omar Sanseviero, Joshua Lochner and Lucain Pouget for help with integration into HuggingFace. We also thank beta testers Javier Ferrando, Oscar Balcells Obeso and others for additional feedback. We are grateful for help and contributions from Phoebe Kirk, Andrew Forbes, Arielle Bier, Aliya Ahmad, Yotam Doron, Ludovic Peran, Anand Rao, Samuel Albanie, Dave Orr, Matt Miller, Alex Turner, Shruti Sheth, Jeremy Sie and Glenn Cameron.

## Author contributions

Tom Lieberum (TL) led the writing of the report, and implementation and running of evaluations. TL also led optimization of SAE training code and fast distributed data loading with significant contributions from Vikrant Varma (VV) and Lewis Smith (LS). Senthooran Rajamanoharan (SR) developed the JumpReLU architecture, led SAE training and significantly contributed to writing and editing the report. SAEs were trained using a codebase that was designed and implemented by TL and VV with significant contributions from Arthur Conmy (AC), which in turn relies on an interpretability codebase written in large part by János Kramár (JK). JK also wrote Mishax, a python library that was used to seamlessly adapt our codebase to the newest Gemma models, which was open-sourced with contribution from Nicolas Sonnerat (NS). AC led the early access and open sourcing of code and weights with significant contribution from LS, in addition to training and evaluating the transcoders and IT SAEs with significant contribution from SR. LS wrote the Gemma Scope tutorial. Neel Nanda (NN) wrote the list of open problems in Appendix A and led coordination with the various stakeholders required to make the launch possible. Anca Dragan (AD), Rohin Shah (RS) and NN provided leadership and advice throughout the project and edited the report.

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

# A  Open problems that Gemma Scope may help tackle

Our main goal in releasing Gemma Scope is to help the broader safety and interpretability communities advance our understanding of interpretability, and how it can be used to make models safer. As a starting point, we provide a list of open problems we would be particularly excited to see progress on, where we believe Gemma Scope may be able to help. Where possible we cite work that may be a helpful starting point, even if it is not tackling exactly the same question.

**Deepening our understanding of SAEs**

1. Exploring the structure and relationships between SAE features, as suggested in Wattenberg and Viégas (2024).
2. Comparisons of residual stream SAE features across layers, e.g. are there persistent features that one can "match up" across adjacent layers?
3. Better understanding the phenomenon of "feature splitting" (Bricken et al., 2023) where high-level features in a small SAE break apart into several finer-grained features in a wider SAE.
4. We know that SAEs introduce error, and completely miss out on some features that are captured by wider SAEs (Templeton et al., 2024; Bussmann et al., 2024). Can we quantify and easily measure "how much" they miss and how much this matters in practice?
5. How are circuits connecting up superposed features represented in the weights? How do models deal with the interference between features (Nanda et al., 2023b)?

**Using SAEs to improve performance on real–world tasks (compared to fair baselines)**

1. Detecting or fixing jailbreaks.
2. Helping find new jailbreaks/red-teaming models (Ziegler et al., 2022).
3. Comparing steering vectors (Turner et al., 2024) to SAE feature steering (Conmy and Nanda, 2024) or clamping (Templeton et al., 2024).
4. Can SAEs be used to improve interpretability techniques, like steering vectors, such as by removing irrelevant features (Conmy and Nanda, 2024)?

**Red-teaming SAEs**

1. Do SAEs really find the "true" concepts in a model?
2. How robust are claims about the interpretability of SAE features (Huang et al., 2023)?
3. Can we find computable, quantitative measures that are a useful proxy for how "interpretable" humans think a feature vector is (Bills et al., 2023)?
4. Can we find the "dark matter" of truly non-linear features?[11]
5. Do SAEs learn spurious compositions of independent features to improve sparsity as has been shown to happen in toy models (Anders et al., 2024), and can we fix this?

**Scalable circuit analysis: What interesting circuits can we find in these models?**
1. What's the learned algorithm for addition (Stolfo et al., 2023) in Gemma 2 2B?
2. How can we practically extend the SAE feature circuit finding algorithm in Marks et al. (2024) to larger models?
3. Can we use SAE-like techniques such as MLP transcoders (Dunefsky et al., 2024) to find input independent, weights-based circuits?

**Using SAEs as a tool to answer existing questions in interpretability**
1. What does finetuning do to a model's internals (Jain et al., 2024)?
2. What is actually going on when a model uses chain of thought?
3. Is in-context learning true learning, or just promoting existing circuits (Hendel et al., 2023; Todd et al., 2024)?
4. Can we find any "macroscopic structure" in language models, e.g. families of features that work together to perform specialised roles, like organs in biological organisms?[12]
5. Does attention head superposition (Jermyn et al., 2023) occur in practice? E.g. are many attention features spread across several heads (Kissane et al., 2024b)?

**Improvements to SAEs**
1. How can SAEs efficiently capture the circular features of Engels et al. (2024)?

---

[11]We distinguish truly non-linear features from low-rank subspaces of linear features as found in Engels et al. (2024).

[12]We know this happens in image models (Voss et al., 2021) but have not seen much evidence in language models. But superposition is incentivized for features that do not co-occur (Gurnee et al., 2023), so specialized macroscopic structure may be a prime candidate to have in superposition. Now we have SAEs, can we find and recover it?

2. How can they deal efficiently with cross-layer superposition, i.e. features produced in superposition by neurons spread across multiple layers?
3. How much can SAEs be quantized without significant performance degradation, both for inference and training?

## B   Standardizing SAE parameters for inference

As described in Section 3, during training, we normalize LM activations and subtract $\mathbf{b}_{\text{dec}}$ from them before passing them to the encoder. However, after training, we reparameterize the Gemma Scope SAEs so that neither of these steps are required during inference.

Let $\mathbf{x}_{\text{raw}}$ be the raw LM activations that we rescale by a scalar constant $C$, i.e. $\mathbf{x} := \mathbf{x}_{\text{raw}}/C$, such that $\mathbb{E}\left[\|\mathbf{x}\|_2^2\right] = 1$. Then, as parameterized during training, the SAE forward pass is defined by

$$\mathbf{f}(\mathbf{x}_{\text{raw}}) := \text{JumpReLU}_{\boldsymbol{\theta}}\left(\mathbf{W}_{\text{enc}}\left(\frac{\mathbf{x}_{\text{raw}}}{C} - \mathbf{b}_{\text{dec}}\right) + \mathbf{b}_{\text{enc}}\right), \tag{5}$$

$$\hat{\mathbf{x}}_{\text{raw}}(\mathbf{f}) := C \cdot \left(\mathbf{W}_{\text{dec}}\mathbf{f} + \mathbf{b}_{\text{dec}}\right). \tag{6}$$

It is straightforward to show that by defining the following rescaled and shifted parameters:

$$\mathbf{b}'_{\text{enc}} := C\,\mathbf{b}_{\text{enc}} - C\,\mathbf{W}_{\text{enc}}\mathbf{b}_{\text{dec}} \tag{7}$$

$$\mathbf{b}'_{\text{dec}} := C\,\mathbf{b}_{\text{dec}} \tag{8}$$

$$\boldsymbol{\theta}' := C\,\boldsymbol{\theta} \tag{9}$$

we can simplify the SAE forward pass (operating on the raw activations $\mathbf{x}_{\text{raw}}$) as follows:

$$\mathbf{f}(\mathbf{x}_{\text{raw}}) = \text{JumpReLU}_{\boldsymbol{\theta}'}\left(\mathbf{W}_{\text{enc}}\mathbf{x}_{\text{raw}} + \mathbf{b}'_{\text{enc}}\right), \tag{10}$$

$$\hat{\mathbf{x}}_{\text{raw}}(\mathbf{f}) = \mathbf{W}_{\text{dec}}\mathbf{f} + \mathbf{b}'_{\text{dec}}. \tag{11}$$

## C   Transcoders

MLP SAEs are trained on the output of MLPs, but we can also replace the whole MLP with a *transcoder* (Dunefsky et al., 2024) for easier circuit analysis. Transcoders are not autoencoders: while SAEs are trained to reconstruct their input, transcoders are trained to approximate the output of MLP layers from the input of the MLP layer. We train one suite of transcoders on Gemma 2B PT, and release these at https://huggingface.co/google/gemma-scope-2b-pt-transcoders.

**Evaluation** We find that transcoders cause a greater increase in loss to the base model relative to the MLP output SAEs (Fig. 6), at a fixed sparsity (L0). This reverses the trend from GPT-2 Small found by Dunefsky et al. (2024). This could be due to a number of factors, such as:

1. Transcoders do not scale to larger models or modern transformer architectures (e.g. Gemma 2 has Gated MLPs unlike GPT-2 Small) as well as SAEs.

2. JumpReLU provides a bigger performance boost to SAEs than to transcoders.

3. Errors in the implementation of transcoders in this work, or in the SAE implementation from Dunefsky et al. (2024).

4. Other training details (not just the JumpReLU architecture) that improve SAEs more than transcoders. Dunefsky et al. (2024) use training methods such as using a low learning rate, differing from SAE research that came out at a similar time to Bricken et al. (2023) such as Rajamanoharan et al. (2024a) and Cunningham et al. (2023). However, Dunefsky et al. (2024) also do not use resampling (Bricken et al., 2023) or an architecture which prevents dead features like more recent SAE research (Rajamanoharan et al., 2024a; Conerly et al., 2024; Gao et al., 2024), which means their results are in a fairly different setting to other SAE research.

**Language model technical details** We fold the pre-MLP RMS norm gain parameters (Zhang and Sennrich (2019), Section 3) into the MLP input matrices, as described in (Gurnee et al. (2024), Appendix A.1) and then train the transcoder on input activations just after the pre-MLP RMSNorm, to reconstruct the MLP sublayer's output as the target activations. To make it easier for Gemma Scope users to apply this change, in Fig. 7 we provide TransformerLens code for loading Gemma 2 2B with this weight folding applied. Fig. 7 also includes an explanation of why only a subset of the weight folding techniques described in Appendix A.1 of Gurnee et al. (2024) can be applied to Gemma 2, due to its architecture.

**Technical details of transcoder training** We train transcoders identically to MLP SAEs except for the following two differences:

1. We do not initialize the encoder kernel $\mathbf{W}_{\text{enc}}$ to the transpose of the decoder kernel $\mathbf{W}_{\text{dec}}$;

2. We do not use a pre-encoder bias, i.e. we do not subtract $\mathbf{b}_{\text{dec}}$ from the input to the transcoder (although we still add $\mathbf{b}_{\text{dec}}$ at the transcoder output).

These two training changes were motivated by the fact that, unlike SAEs, the input and outputs spaces for transcoders are not identical. To spell out how we apply normalization: we divide the input and target activations by the root mean square of the input activations. Since the input activations all have norm $\sqrt{d_{\text{model}}}$ due to RMSNorm, this means we divide input and output activations by $\sqrt{d_{\text{model}}}$.

# D  Additional evaluation results

## D.1  Sparsity-fidelity tradeoff

Fig. 11 illustrates the trade off between fidelity as measured by fraction of variance unexplained (FVU) against sparsity for layer 12 Gemma 2 2B and layer 20 Gemma 2 9B SAEs.

Fig. 12 shows the sparsity-fidelity trade off for the 131K-width residual stream SAEs trained on Gemma 2 27B after layers 10, 22 and 34 that we include as part of this release.

Fig. 15 and Fig. 16 show fidelity versus sparsity curves for more layers (approximately evenly spaced) and all sites of Gemma 2 2B and Gemma 2 9B, demonstrating consistent and smoothly variance performance throughout these models.

## D.2  Impact of sequence position

**Methodology** Prior research has shown that language models tend to have lower loss on later token positions (Olsson et al., 2022). It is thus natural to ask how an SAE's performance changes over the length of a sequence. Similar to Section 4.1, we track reconstruction loss and delta loss for various sparsity settings, however this time we do not aggregate over the sequence position. Again, we mask out special tokens.

**Result** Fig. 8 shows how reconstruction loss varies by position for 131K-width SAEs trained on the middle-layer of Gemma 2 9B. Reconstruction loss increases rapidly from close to zero over the first few tokens. The loss monotonically increases by position for attention SAEs, although it is essentially flat after 100 tokens. For MLP SAEs, the loss peaks at around the tenth token before gradually declining slightly. We speculate that this is because attention is most useful when tracking long-range dependencies in text, which matters most when there is significant prior context to draw

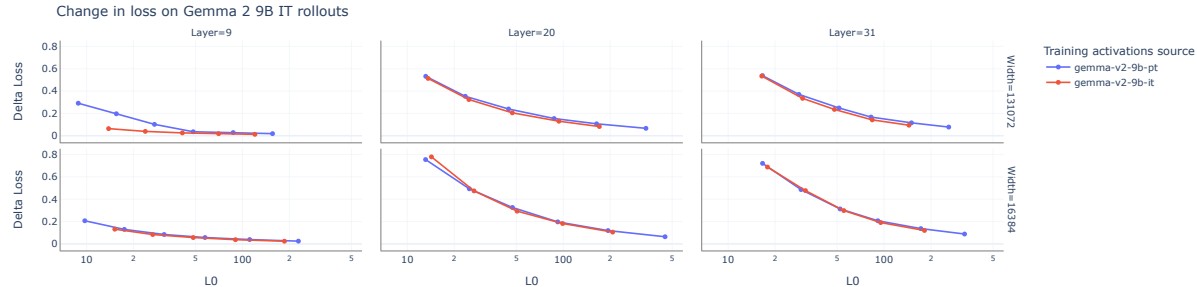

Figure 5: Change in loss when splicing in SAEs trained on Gemma 2 9B (base and IT) to reconstruct the activations generated with Gemma 2 9B IT on rollouts.

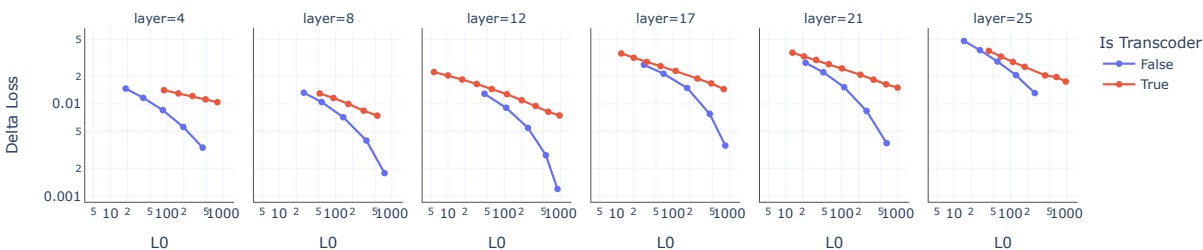

Figure 6: Transcoders trained to reconstruct MLP output from the MLP input cause a greater increase in loss compared to the vanilla model when compared with an MLP output SAE. The sites are (the MLP sub-) layers throughout Gemma 2B PT.

```python
import transformer_lens # pip install transformer-lens

model = transformer_lens.HookedTransformer.from_pretrained(
    "google/gemma-2-2b",
    # In Gemma 2, only the pre-MLP, pre-attention and final RMSNorms can
    # be folded in (post-attention and post-MLP RMSNorms cannot be folded in):
    fold_ln=True,
    # Only valid for models with LayerNorm, not RMSNorm:
    center_writing_weights=False,
    # These model use logits soft-capping, meaning we can't center unembed:
    center_unembed=False,
)
```

Figure 7: Code for loading Gemma 2B in TransformerLens (Nanda and Bloom, 2022) to use this with our Transcoders.

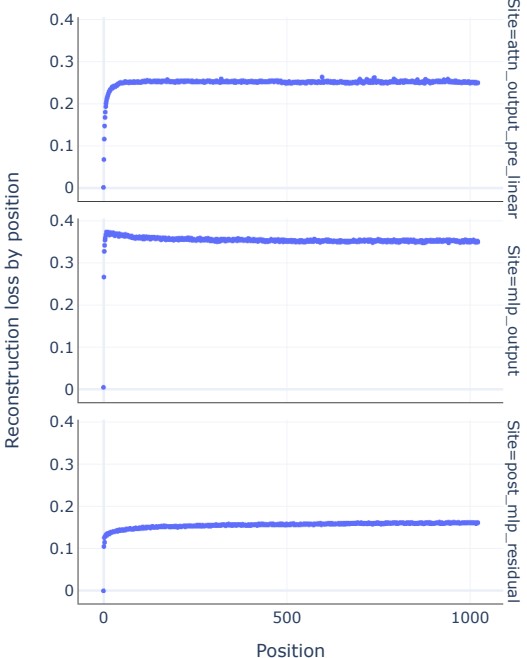

Figure 8: Reconstruction loss by sequence position for Gemma 2 9B middle-layer 131K-width SAEs with $\lambda = 10^{-3}$.

from, while MLP layers do a lot of local processing, like detecting n-grams (Gurnee et al., 2023), that does not need much context. Like attention SAEs, residual stream SAEs' loss monotonically increases, plateauing more gradually. Curves for other models, layers, widths and sparsity coefficients were found to be qualitatively similar.

Fig. 13 shows how delta LM loss varies by sequence position. The high level of noise in the delta loss measurements makes it difficult to robustly measure the effect of position, however there are signs that this too is slightly lower for the first few tokens, particularly for residual stream SAEs.

### D.3   Pile subsets

**Methodology**   We perform the sparsity-fidelity evaluation from Section 4.1 on different validation subsets of The Pile (Gao et al., 2020), to gauge whether SAEs struggle with a particular type of data.[13]

**Results**   In Fig. 9 we show delta loss by subset. Of the studied subsets, SAEs perform best on Deep-Mind mathematics (Saxton et al., 2019). Possibly

---

[13]Note that this is a different dataset to the dataset used to train the Gemma Scope SAEs.

this is due to the especially formulaic nature of the data. SAEs perform worst on Europarl (Koehn, 2005), a multilingual dataset. We conjecture that this is due to the Gemma 1 pretraining data, which was used to train the SAEs, containing predominantly English text.

### D.4   Impact of low precision inference

We train all SAEs in 32-bit floating point precision. It is common to make model inference less memory and compute intensive by reducing the precision at inference time. This is particularly important for applications like circuit analysis, where users may wish to splice several SAEs into a language model simultaneously. Fig. 10 compares fidelity-versus-sparsity curves computed using `float32` SAE and LM weights versus the same curves computed using `bfloat16` SAE and LM weights, suggesting there is negligible impact in switching to `bfloat16` for inference.

### D.5   Uniformity of active latent importance

**Methodology**   Conventionally, sparsity of SAE latent activations is measured as the L0 norm of the latent activations. Olah et al. (2024) suggest to train SAEs to have low L1 activation of attribution-weighted latent activations, taking into account that some latents may be more important than others. We repurpose their loss function as a metric for our SAEs, which were trained penalising activation sparsity as normal. As in Rajamanoharan et al. (2024b), we define the attribution-weighted latent activation vector $\mathbf{y}$ as

$$\mathbf{y} := \mathbf{f}(\mathbf{x}) \odot \mathbf{W}_{\text{dec}}^T \nabla_{\mathbf{x}} \mathcal{L}, \qquad (12)$$

where we choose the mean-centered logit of the correct next token as the loss function $\mathcal{L}$.

We then normalize the magnitudes of the entries of $\mathbf{y}$ to obtain a probability distribution $p \equiv p(\mathbf{y})$. We can measure how far this distribution diverges from a uniform distribution $u$ over active latents via the KL divergence

$$\mathbf{D}_{\text{KL}}(p\|u) = \log \|\mathbf{y}\|_0 - \mathbf{S}(p), \qquad (13)$$

with the entropy $\mathbf{S}(p)$. Note that $0 \leq \mathbf{D}_{\text{KL}}(p\|u) \leq \log \|\mathbf{y}\|_0$. Exponentiating the negative KL divergence gives a new measure $r_{L0}$

$$r_{L0} := e^{-\mathbf{D}_{\text{KL}}(p\|u)} = \frac{e^{\mathbf{S}(p)}}{\|\mathbf{y}\|_0}, \qquad (14)$$

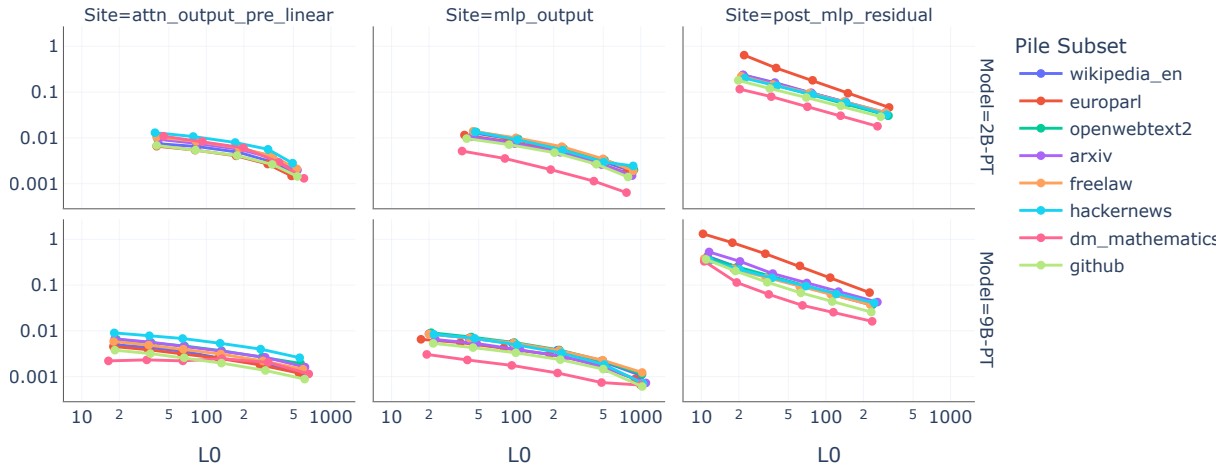

Figure 9: Delta loss per pile subset (65K for 2B, 131K for 9B).

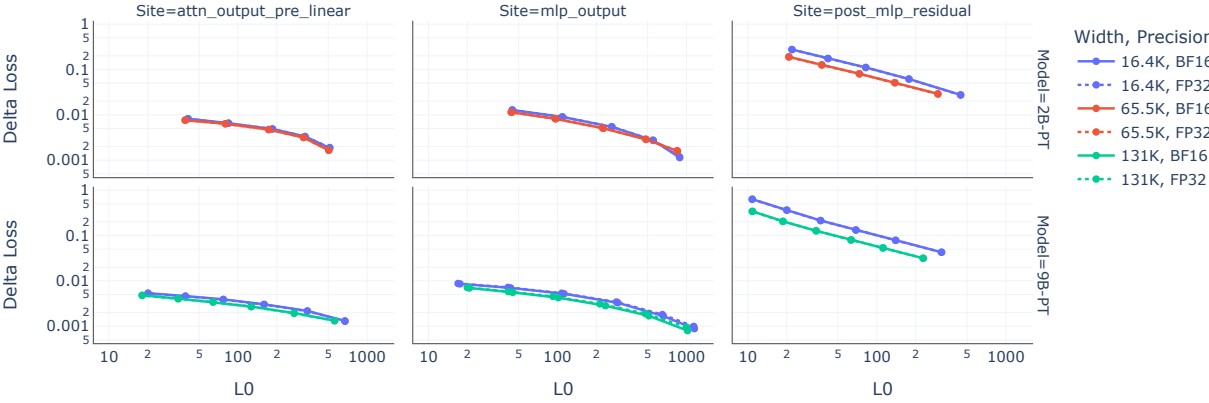

Figure 10: Delta loss versus sparsity computed using either `float32` or `bfloat16` SAE and language model weights.

with $\frac{1}{\|\mathbf{y}\|_0} \leq r_{L0} \leq 1$. Note that since $e^{\mathbf{S}}$ can be interpreted as the effective number of active elements, $r_{L0}$ is the ratio of the effective number of active latents (after re-weighting) to the total number of active latents, which we call the 'Uniformity of Active Latent Importance'.

**Results**   In Fig. 14 we show $r_{L0}$ on middle layer SAEs. In line with Rajamanoharan et al. (2024b), we find that the attributed effect becomes more diffuse as more latents are active. This effect is most pronounced for residual stream SAEs, and seems to be independent of language model size and number of SAE latents.

### D.6    Additional Gemma 2 IT evaluation results

In this sub-appendix, we provide further evaluations of SAEs on the activations of IT models, continuing Section 4.4.

As mentioned in Section 4.4, we find in Fig. 19 that PT SAEs achieve reasonable FVU on rollouts, but the gap between PT and IT SAEs is larger than in the change in loss in the main text (Fig. 5).

In Fig. 17 we evaluate the FVU on the user prompt and model prefix (not the rollout). In Fig. 18 we evaluate the change in loss (delta loss) on the user prompts, and surprisingly find that splicing in the base model SAE can reduce the loss in expectation in some cases. Our explanation for this result is that post-training does not train models to predict user queries (only predict high-preference model rollouts) and therefore the model is not incentivised to have good predictive loss by default on the user prompt.

While we do not train IT SAEs on Gemma 2 2B, we find that the base SAEs transfer well as measured by FVU in Fig. 20.

Finally, we do not find evidence that rescaling IT activations to have same norm in expectation to the pretraining activations is beneficial (Fig. 21). The trend for individual SAEs in this plot is that their L0 decreases but the Pareto frontier is very slightly worse. This is consistent with prior observations that SAEs are surprisingly adaptable to different L0s (Smith, 2024; Gao et al., 2024).

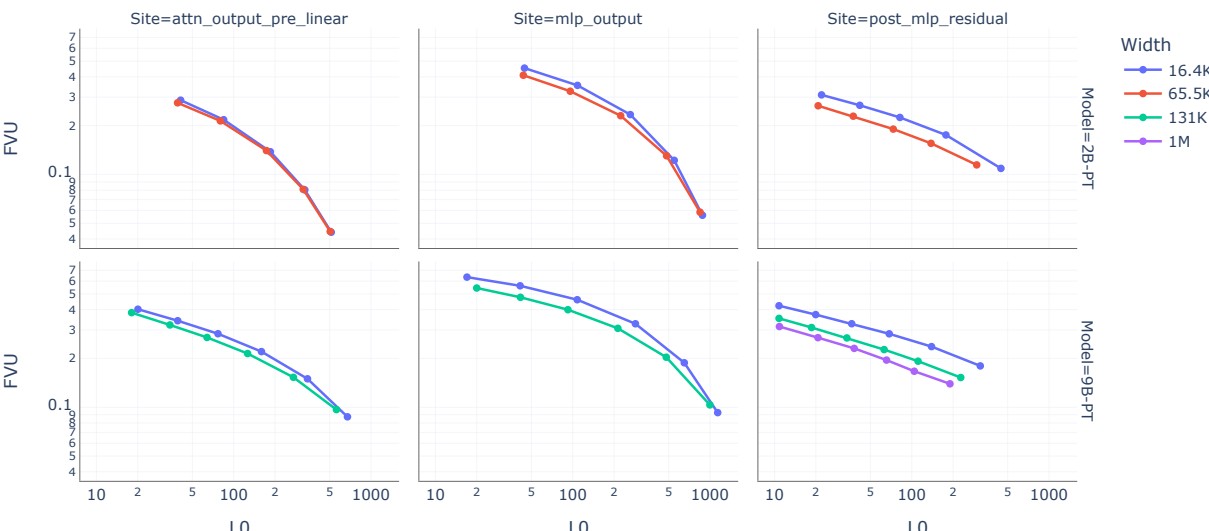

Figure 11: Sparsity-fidelity trade-off for middle-layer Gemma 2 2B and 9B SAEs using fraction of variance unexplained (FVU) as the measure of reconstruction fidelity.

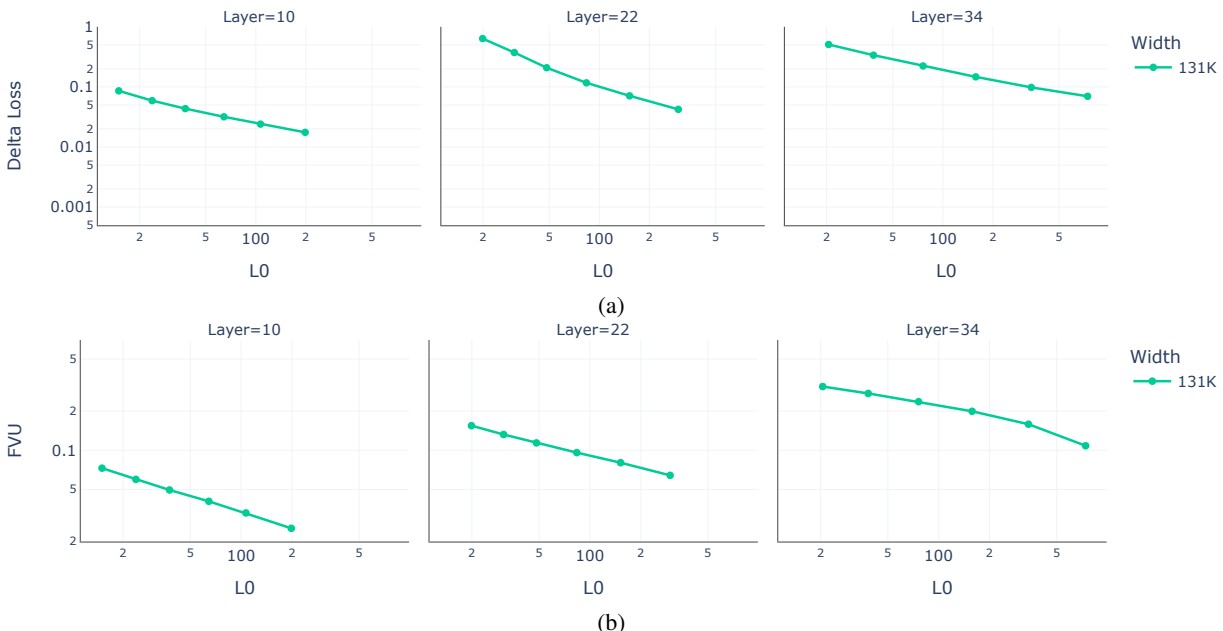

Figure 12: Sparsity-fidelity trade-off for Gemma 2 27B SAEs using (a) delta LM loss and (b) as measures of reconstruction fidelity.

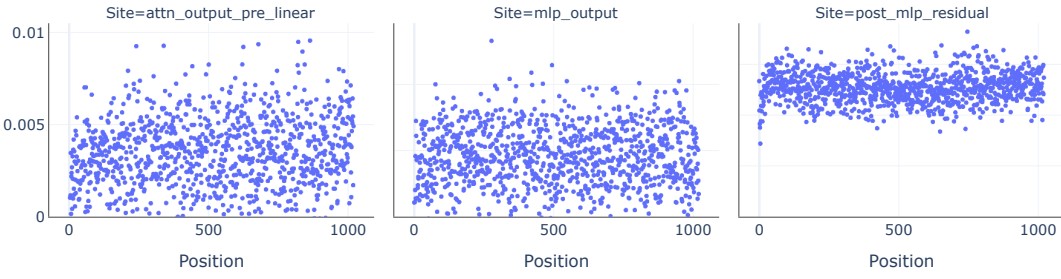

Figure 13: Delta loss by sequence position for Gemma 2 9B middle-layer 131K-width SAEs with $\lambda = 10^{-3}$.

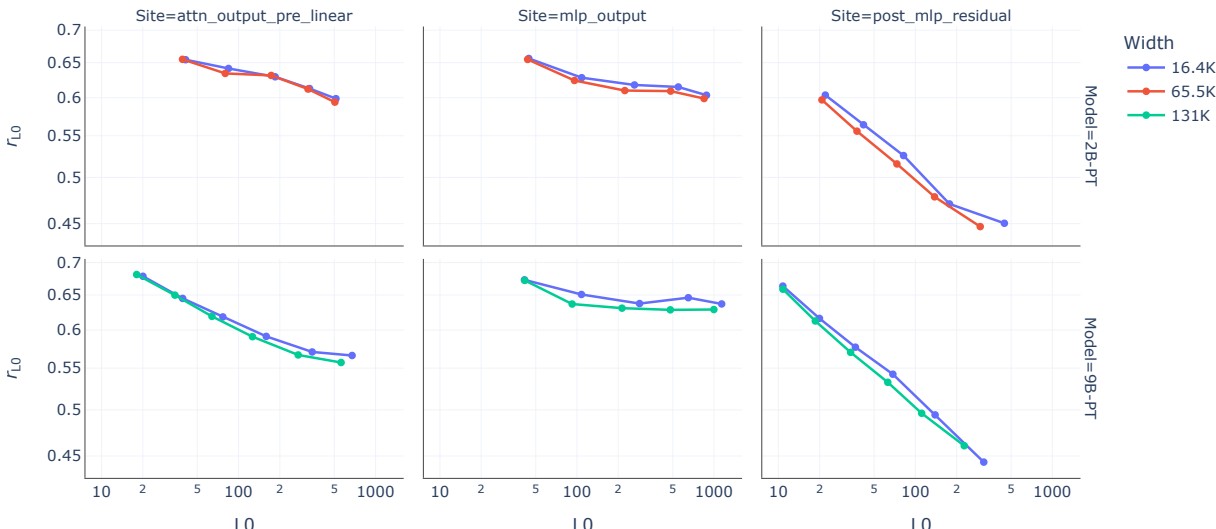

Figure 14: Uniformity of active latent importance for the middle layer SAEs.

Figure 15: Sparsity-fidelity trade-off across multiple layers of Gemma 2 2B, approximately evenly spaced. (Note Gemma 2 2B has 26 layers.)

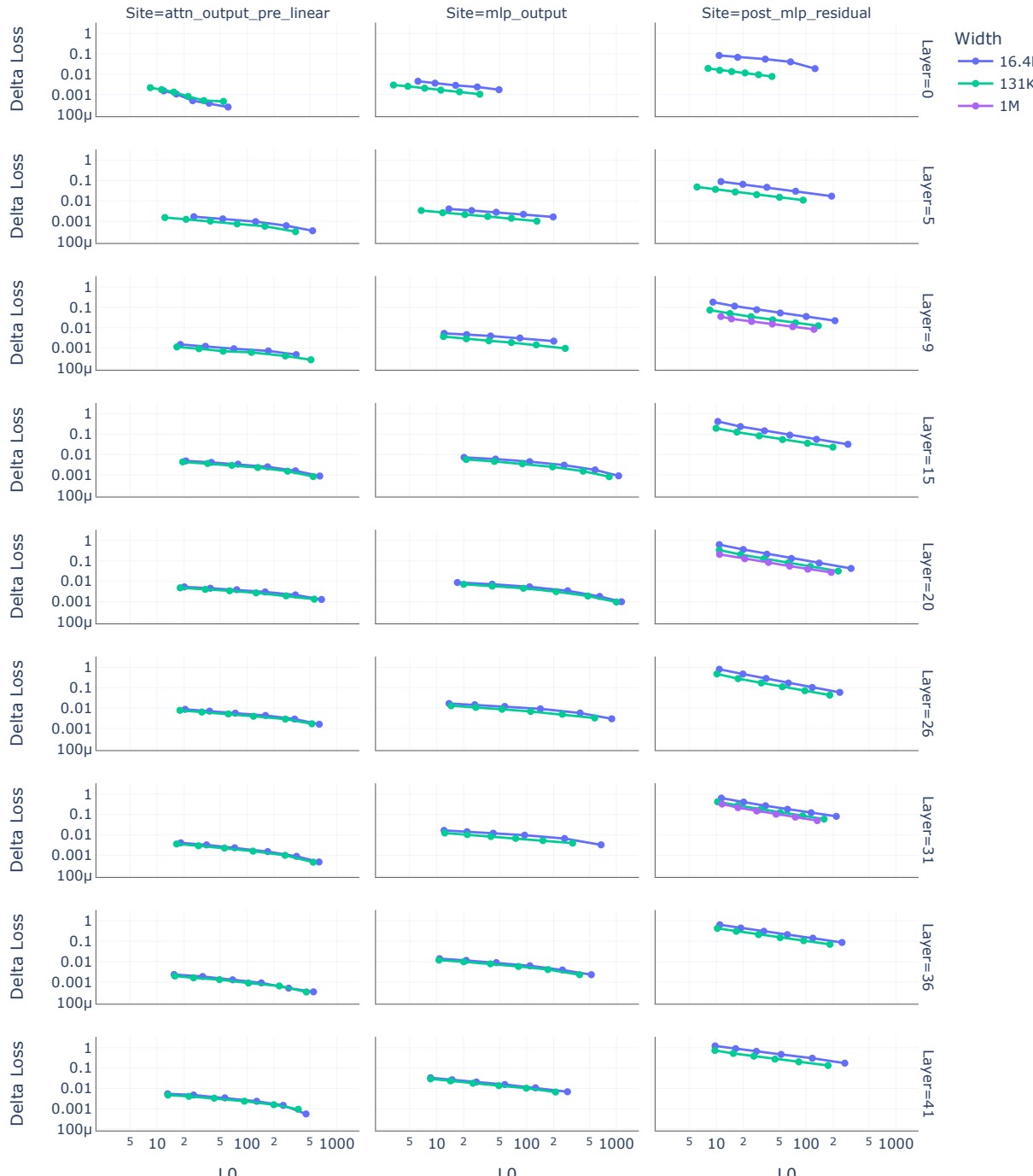

Figure 16: Sparsity-fidelity trade-off across multiple layers of Gemma 2 9B, approximately evenly spaced. (Note Gemma 2 2B has 42 layers.)

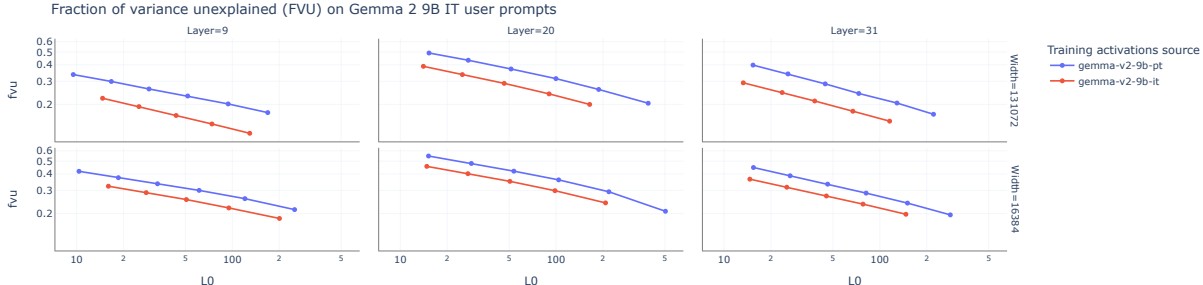

Figure 17: Fraction of variance unexplained when using SAEs trained on Gemma 2 9B (base and IT) to reconstruct the activations generated with Gemma 2 9B IT on user prompts.

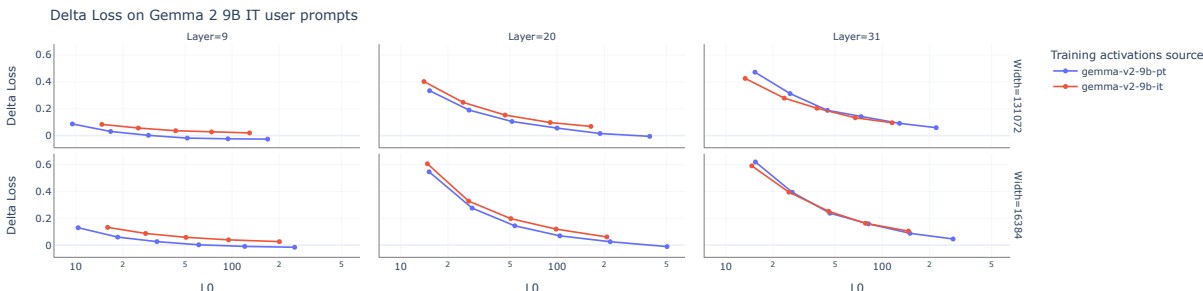

Figure 18: Change in loss when splicing in SAEs trained on Gemma 2 9B (base and IT) to reconstruct the activations generated with Gemma 2 9B IT on user prompts.

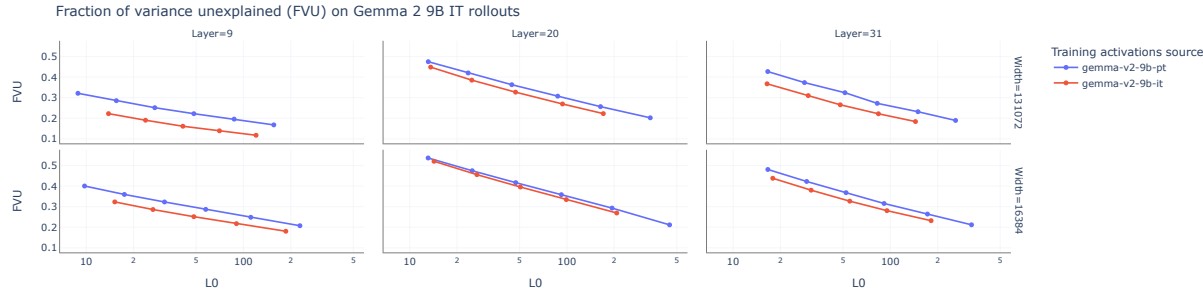

Figure 19: Fraction of variance unexplained when using SAEs trained on Gemma 2 9B (base and IT) to reconstruct the activations generated with Gemma 2 9B IT on rollouts.

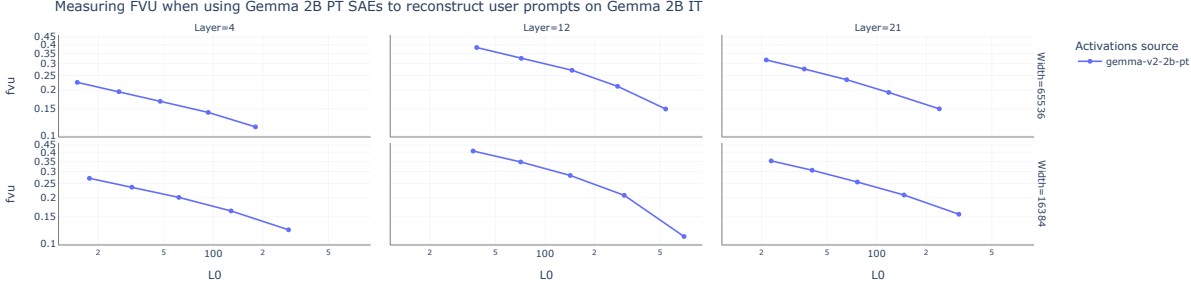

Figure 20: Fraction of variance unexplained when using SAEs trained on Gemma 2 2B PT to reconstruct the activations generated with Gemma 2 2B IT on user prompts.

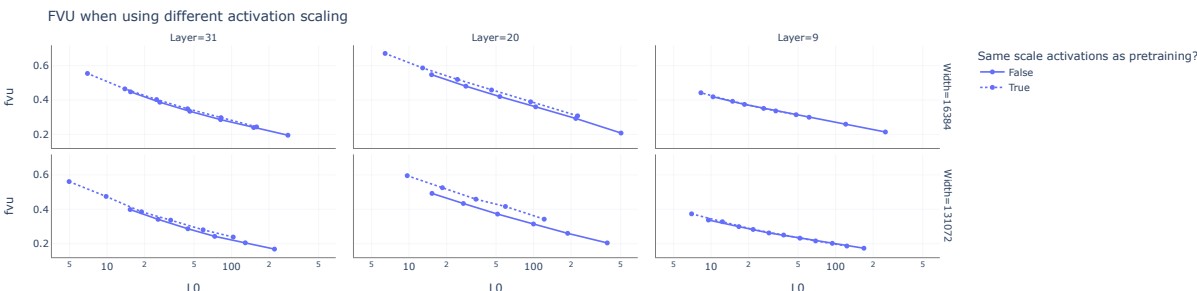

Figure 21: Fraction of variance unexplained when using SAEs trained on Gemma 2 9B PT to reconstruct the activations generated with Gemma 2 9B IT on rollouts, including when rescaling the IT activations to have the same norm (in expectation) as the pretraining activations.