# OpenReview forum: "Gemma Scope: Open Sparse Autoencoders Everywhere All At Once on Gemma 2"
_EMNLP/2024/Workshop/BlackBoxNLP — BlackboxNLP 2024_

### Official Review · Reviewer_3zBf · 2024-08-28

**Overall Assessment:** 4
**Confidence:** 3

**Best Paper:**

2

**Best Paper Justification:**

the resources released with the paper are quite unique due to their scale; hard for me to judge how impactful it will be, because this stands and falls with the success of SAEs in general, but it certainly has the potential of large impact.

**Comments Questions Suggestions And Typos:**

-

**Paper Summary:**

This paper aims to accelerate interpretability research using sparse autoencoders (SAE), which suffer from high training cost and a lack of openly released SAEs for modern models. To support future research, authors release SAEs of varying sparsity for each (sub)layer of Gemma 2 9B and 2B, as well as selected layers of Gemma 2 27B.

The paper goes into detail about training details and engineering challenges addressed during this project, and has some evaluation results, although the main success criterion is to what extent future interpretability research will be able to benefit from the SAEs released.

**Summary Of Strengths:**

- the paper presents the results of a massive effort, both on the level of engineering and computation, and the number of SAEs produced is similarly impressive in scale.
- the paper has the potential to accelerate research on SAEs and make this research direction accessible to groups who lack computational power to train their own SAEs on this scale.
- the paper is dense in technical details, but still well-organized and readable.

**Summary Of Weaknesses:**

- massive scale is not only a strength, but brings environmental costs. I hope the resources put into this don't go to waste, for example because the particular flavour of SAE used becomes obsolete. Authors say that they used "over 20% of the training compute of GPT-3", but applications of SAEs are much more narrow than the LLMs themselves.

---

### Official Review · Reviewer_fGTE · 2024-08-29

**Overall Assessment:** 5
**Confidence:** 4

**Best Paper:**

2

**Best Paper Justification:**

The release of these SAEs is of significant value to the interpretability community and could enable a plethora of follow-up work.

**Comments Questions Suggestions And Typos:**

- The reference to Nora Belrose's SAEs is titled "Untitled" (p. 9).

**Paper Summary:**

This technical report describes the training procedures used to train a comprehensive suite of JumpReLU SAEs on Gemma 2 (Instruct). Specifically, the authors trained multiple suites: a comprehensive suite of SAEs on every layer and sublayer of Gemma 9B and 2B, a selection of layers of Gemma 27B, a suite of transcoders, a suite of SAEs with multiple widths, and a selection of layers on Gemma 2 9B Instruct (over 2,000 SAEs in total). Finally, the authors evaluate and compare the quality of their SAEs on standard metrics, and propose a set of research questions this suite of SAEs could help to address.

**Summary Of Strengths:**

- Training a comprehensive suite of SAEs on models of this size is a significant engineering effort and very compute intensive. As a result, training these SAEs and making them available is very valuable for the community as it enables researchers to explore research questions they would otherwise not be able to.
- Gemma 2 and JumpReLU SAEs are both state-of-the-art architectures.
- The quality of the SAEs appears to be very good.

**Summary Of Weaknesses:**

NA

---

### Decision · Program_Chairs · 2024-09-19

**Decision:**

Accept

**Comment:**

Reviewers appreciated the contribution of a massive suite of SAEs trained at large scale on SOTA models, as well as the technical details. One reviewer raised a concern about environmental costs in case the particular type of SAEs becomes obsolete. It would be good to discuss this in the camera ready.